# Cu(II) Binding Increases the Soluble Toxicity of Amyloidogenic Light Chains

**DOI:** 10.3390/ijms23020950

**Published:** 2022-01-16

**Authors:** Rosaria Russo, Margherita Romeo, Tim Schulte, Martina Maritan, Luca Oberti, Maria Monica Barzago, Alberto Barbiroli, Carlo Pappone, Luigi Anastasia, Giovanni Palladini, Luisa Diomede, Stefano Ricagno

**Affiliations:** 1Dipartimento di Fisiopatologia Medico-Chirurgica e Dei Trapianti, Università Degli Studi di Milano, 20090 Segrate, Italy; rosaria.russo@unimi.it; 2Dipartimento di Biochimica e Farmacologia Molecolare, Istituto di Ricerche Farmacologiche Mario Negri IRCCS, 20156 Milan, Italy; Margherita.Romeo@iuf-duesseldorf.de (M.R.); mariamonica.barzago@marionegri.it (M.M.B.); 3Institute of Molecular and Translational Cardiology, IRCCS Policlinico San Donato, 20097 Milan, Italy; TimPaul.Schulte@grupposandonato.it (T.S.); carlo.pappone@af-ablation.org (C.P.); Anastasia.luigi@hsr.it (L.A.); 4Dipartimento di Bioscienze, Università Degli Studi di Milano, 20133 Milano, Italy; mmaritan@scripps.edu (M.M.); dr.luca.oberti@gmail.com (L.O.); 5Dipartimento di Scienze per gli Alimenti, La Nutrizione e L’Ambiente, Università Degli Studi di Milano, 20133 Milan, Italy; alberto.barbiroli@unimi.it; 6Arrhythmia and Electrophysiology Department, IRCCS Policlinico San Donato, San Donato, 20097 Milan, Italy; 7Faculty of Medicine and Surgery, Vita-Salute San Raffaele University, 20132 Milan, Italy; 8Amyloidosis Treatment and Research Center, Fondazione IRCCS Policlinico San Matteo, Università Degli Studi di Pavia, 27100 Pavia, Italy; giovanni.palladini@unipv.it

**Keywords:** light chain amyloidosis, soluble toxicity, protein aggregation, copper ions, copper binding

## Abstract

Light chain amyloidosis (AL) is caused by the aberrant overproduction of immunoglobulin light chains (LCs). The resulting abnormally high LC concentrations in blood lead to deposit formation in the heart and other target organs. Organ damage is caused not only by the accumulation of bulky amyloid deposits, but extensive clinical data indicate that circulating soluble LCs also exert cardiotoxic effects. The nematode *C. elegans* has been validated to recapitulate LC soluble toxicity in vivo, and in such a model a role for copper ions in increasing LC soluble toxicity has been reported. Here, we applied microscale thermophoresis, isothermal calorimetry and thermal melting to demonstrate the specific binding of Cu^2+^ to the variable domain of amyloidogenic H7 with a sub-micromolar affinity. Histidine residues present in the LC sequence are not involved in the binding, and yet their mutation to Ala reduces the soluble toxicity of H7. Copper ions bind to and destabilize the variable domains and induce a limited stabilization in this domain. In summary, the data reported here, elucidate the biochemical bases of the Cu^2+^-induced toxicity; moreover, they also show that copper binding is just one of the several biochemical traits contributing to LC soluble in vivo toxicity.

## 1. Introduction

Systemic amyloid diseases are characterized by the aberrant accumulation of amyloid fibrils in the extracellular space of target organs [1,2]. Several organs can be involved, and amyloid accumulation is associated with organ disfunction. Defining the molecular features that predispose to amyloid formation, understanding the molecular bases of proteotoxicity and the mechanisms causing organ dysfunction are crucial steps for deciphering and treating these pathologic conditions.

Protein misfolding and tissue damage are intuitively related processes, but the link between them remains unclear in many cases. In most amyloid-related diseases, the presence of amyloid deposits *per se* is not considered sufficient to explain the clinical phenotype [3,4,5,6,7,8]. In central amyloidosis monomeric and fibrillar species do not display any toxicity in vitro or in vivo, while highly toxic oligomeric species, characterized by a loose structure and a high level of hydrophobicity, have been identified. In systemic immunoglobulin light chain amyloidosis (AL), the bulky extracellular deposits often alter the macro- and micro-architecture of tissues affecting organ functionality, but the direct cytotoxicity of soluble and circulating protein species has also been clearly demonstrated [9,10,11,12,13,14,15,16,17].

AL is the most common form of systemic amyloidosis: amyloidogenic misfolding-prone monoclonal immunoglobulin light chains (LCs) are produced in excess by a bone marrow plasma cell clone and transported to the target organs through the bloodstream, where they ultimately aggregate as large amyloid deposits [18]. AL is a polymorphic disease and very heterogenous from a clinical point of view, with most patients showing multi-organ involvement. Heart involvement is particularly frequent (~75% of cases) and dramatically worsens patients’ prognosis, but renal deposition is also common (~65% of cases) [19,20,21,22]. The molecular details of such an organ-selective involvement are not understood, but the extremely high sequence variability among LCs caused by genetic rearrangement and somatic hypermutation, probably plays a role.

The heterogeneity of monoclonal LCs complicates efforts to associate sequence determinants with aggregation propensity. Growing experimental evidence indicates that specific biochemical properties play a relevant role or at least strongly correlate with LCs amyloidogenicity. Thermodynamic and kinetic stability are lower in amyloidogenic LCs compared to control non-amyloidogenic LCs, whereas protein flexibility and dynamics are typically increased in AL LCs compared to non-AL ones [15,23,24,25,26,27,28,29,30,31,32,33,34,35]. Specific LC germlines are overrepresented among AL sequences, and recent algorithms tend to predict more accurately the AL propensity of LC sequences [36]. The biochemical properties and sequence determinants of LCs aggregation need to be fully clarified.

The structure of native LCs is highly conserved: they are dimeric and consist of two immunoglobulin domains, the variable domain (V_L_) and the constant domain (C_L_) (Figure 1A) [28]. Instead structural analysis of LCs fibrils depicts a very diverse scenario. The four available Cryo-EM structures of the ex vivo fibrils from AL patients show independent assemblies [37,38,39,40]. The ex vivo amyloid deposits show a large group of proteoforms caused by proteolytic events [41,42,43]; whether such proteolysis is a trigger for LCs aggregation or is the result of post-aggregation processes is still under debate [30,31,43,44].

A growing body of clinical and experimental data shows that the toxic effects directly caused by soluble LC species are extremely relevant for the organ damage observed in AL patients [5,11,16,25,45]. Indeed, clinical observations indicate the rapid amelioration of cardiac dysfunction biomarkers upon reduction of the circulating pathogenic LCs with therapy [4]. Such observations can be recapitulated in the nematode *C. elegans*, an established experimental model of LC cardiotoxicity in vivo. In *C. elegans*, the administration of cardiotropic LCs caused a profound functional and structural damage to the pharynx, which is considered an “ancestral heart.” Such damage is associated with the production of reactive oxygen species (ROSs), which cause mitochondrial structural alteration and injury [10,11,25]. Intriguingly, non-amyloidogenic LCs from multiple myeloma patients (M-LC), which produced negligible amount of ROS, were non-toxic [10,11].

The biophysical stabilization of cardiotropic LCs significantly reduces their soluble toxicity in vitro in cardiac cells and in vivo in *C. elegans*, suggesting that fold instability and protein flexibility may play a role in both aggregation propensity and soluble toxicity [25]. The specific propensity of cardiotropic LCs to generate ROS has been reported to be related to their ability to interact with metal ions, particularly copper, known to modulate the polymerization and toxicity of different amyloidogenic proteins. In vivo observations also indicated that the addition of copper divalent ions to cardiotropic LCs, but not non-amyloidogenic LCs, resulted in a distinct increase in the recorded protein in vivo toxicity [11]. Importantly, metal chelators restored the correct cellular copper levels, counteracted ROS generation and the toxicity of cardiotropic LCs.

In this study we applied biophysical microscale thermophoresis (MST), isothermal microcalorimetry (ITC) and thermal melting assays, and we show that copper specifically binds to the V_L_ of the cardiotropic H7 with a low micromolar affinity. Ala-substitution of two histidine residues in the constant domain of H7 (H7-H188A/H197A mutant) does not alter copper binding or impair ROS generation in vitro. However, the His-to-Ala mutations destabilized the constant domain and also significantly reduced the toxicity of the H7 mutant. Our data indicate that copper binding is a relevant player in determining H7 soluble toxicity, but its binding is not sufficient to completely explain the observed toxic effects, which probably rely on a complex interplay between different molecular properties.

## 2. Results

### 2.1. Copper Binds to H7 with Sub-Micromolar Affinity

We have recently shown that the incubation of the amyloidogenic and cardiotropic LC, H7, with Cu^2+^ resulted in an increase in the protein proteotoxicity in vivo; this phenomenon was specific for copper as other divalent ions did not trigger the same effect [11]. Conversely, the presence of Cu^2+^ did not increase the toxicity of non-amyloidogenic and non-toxic LCs [11]. Herein, we report the application of microscale thermophoresis (MST), isothermal calorimetry (ITC) and thermal melting to demonstrate that Cu^2+^ binds specifically to the V_L_ of H7 with sub-micromolar affinity.

In MST, the titration of copper to fluorescently labeled H7, yielded concentration-dependent upward-shifted thermophoresis profiles (Figure 1C). Plotting the derived ΔF_norm_ values against the Cu^2+^ concentration yields a sigmoidal binding curve with an amplitude value of 15 ‰ and a fitted *K*_D_-value of 750 ± 60 nM (Figure 1C). Other mono- or bi-valent cations (Ca^2+^, Fe^2+^, Mg^2+^, Na^+^, Zn^2+^), did not alter the thermophoretic movement of H7 (Figure 1C and Appendix A). The titration of Cu^2+^ to the non-amyloidogenic LC M7, yielded a binding curve with a similar amplitude but a >50 times reduced affinity of 50 ± 15 μM (Figure 1C). Such binding affinities were confirmed using the orthogonal biophysical method, ITC (Figure 1D). The injection of Cu^2+^ to H7 yielded exothermic spikes that returned to baseline after the 12th injection (Figure 1D). The binding isotherm obtained from four experiments was fit globally to yield the *K*_D_ and binding enthalpy (ΔH) values of 900 ± 100 nM and −8.7  ± 0.1 kcal/mole, respectively (Figure 1D). As a negative control experiment, the injection of Cu^2+^ to H7 in the presence of molar excess of EDTA abolished binding due to(EDTA: Cu^2+^) chelation (Figure 1D). The injection of Cu^2+^ to M7 yielded exothermic spikes at approximately half the amplitude compared to H7 (Figure 1D). The estimated *K*_D_ and ΔH values derived from the two experiments were 16 ± 6 μM and 6 ±  1 kcal/mole, respectively. It should be noted that higher M7 and Cu^2+^ concentrations are required for more accurate values. However, taken together with the MST results, we conclude a 20–50 times reduced affinity value for the binding of copper to M7.

In order to assess the effect of Cu^2+^ on H7 stability, temperature ramps monitored by nano differential scanning fluorimetry (nanoDSF) were performed on H7, with and without Cu^2+^ and in presence of EDTA (Figure 1E). Temperature ramps without copper and in the presence of EDTA are well superimposable and show two-step unfolding corresponding to the unfolding of the V_L_ domain at 45 °C and then the C_L_ domain at 57 °C (Figure 1E). It should be noted that a mutation-induced shift of the second T_m_ value in melting curves of the H7-H188A/H197A, strongly suggest the association of the first and second Tm values with the V_L_ and C_L_ domains, respectively (Figure 2C and explained below). The destabilization of C_L_ yielded a melting curve with a single apparent inflection point at 49 °C, indicating more cooperative unfolding of H7-H188A/H197A. Interestingly, H7 titration with Cu^2+^ shifted the first Tm value from 45 °C to 48.3 °C, thus revealing that copper binding induced the stabilization of the V_L_ domain.

### 2.2. H7-H188A/H197A Mutant Binds Cu^2+^ and Produces H_2_O_2_ Similar to Wild-Type H7

With the exception of a few examples, copper binding to proteins is dominated by three ligand types: the side chains of methionine, cysteine and histidine [46]. H7 lacks methionine residues, and four out of five cysteine residues form non-solvent exposed intramolecular disulphide bonds, the fifth forms an intermolecular disulphide bond between the C-terminal regions of the C_L_ domains [28]. The fifth cysteine residue is localized close to the C-*terminus* of the C_L_ domain, rarely resolved in LC crystal structures due to conformational flexibility and is thus unlikely a part of a potential Cu^2+^ coordination site. Therefore, we focused on the two histidine residues present in the H7 sequence, H188 and H197, both localized in the C_L_ domain (Figure 1A).

The substitution of both histidine residues by alanine in the H7-H188A/H197A mutant resulted in SEC and CD spectral profiles similar to the H7 wild-type, suggesting a preserved overall structure of the mutant (Figure 2A,B). However, comparative analysis of the thermal denaturation profiles revealed a sharper transition of the temperature ramp CD profile of H7-H188A/H197A compared to H7, indicating the mutation-induced destabilization of the mutant (Figure 2C). The broad transition of H7 unfolding exhibits two midpoint temperatures of 44 °C and 57 °C that are apparent in CD temperature ramps (Figure 2C). Two independent midpoints were also present and clearly resolved in nanoDSF datasets collected prior and after ITC titrations (Figure 1E). The H188A/H197A mutations located within the C_L_ domain caused a substantial shift of the second T_m_ value from 57 °C to 51.5 °C, suggesting a mutation-induced destabilization of the C_L_ domain and resulting in a more cooperative unfolding.

ITC experiments show that the titration of Cu^2+^ to H7-H188A/H197A yielded *K*_D_ and ΔH values of 900 ±  300 nM and −7.6 ±  0.8 kcal/mole (Figure 1D), which are comparable to those of H7. This observation indicates that histidine residues are not involved in copper binding and that Cu^2+^ is likely bound to the V_L_ domain rather than to the mutated C_L_ domain.

The addition of copper to amyloidogenic LCs facilitates the production of H_2_O_2_, while this was not observed for non-amyloidogenic LCs [11]. Thus, the similar binding affinities of H7 and H7-H188A/H197A for copper ions translated in a comparable capacity of the two proteins to generate oxygen radicals, here measured as the level of H_2_O_2_ produced overtime (Figure 2D). M7, which poorly binds copper ions, generates a much lower amount of oxygen radicals (Figure 2D).

### 2.3. H7-H188A/H197A Mutant Is Less Toxic In Vivo

Finally, H7-H188A/H197A was administered to *C. elegans* and its toxicity was assessed as the reduction in the pharyngeal pumping rate. Surprisingly, H7-H188A/H197A resulted to be significantly less toxic than H7 (Figure 3A), with an IC_50_ 1.6-fold lower (IC_50_: 28.9 µg/mL and 46.8 µg/mL for H7-H188A/H197A and H7, respectively; *p* < 0.01, Student’s *t*-test). The non-amyloidogenic M7 protein, here employed as a negative control, did not impair the pharyngeal function of worms, according to previous data (Appendix A). The addition of copper worsened the pharyngeal toxicity caused by H7, but not the one by H7-H188A/H197A or of M7 (Figure 3B).

## 3. Discussion

Previous in vivo data made a strong case for the toxic effect of Cu^2+^ when delivered by soluble amyloidogenic LCs [11]. In this work, we applied MST, ITC and biophysical protein melting to demonstrate the low micromolar affinity binding of Cu^2+^ to the V_L_ domain of H7. In keeping with the in vivo data, we showed that only copper ions specifically bind to LCs, in accordance with our previous observation that H7 toxicity was not affected by iron and zinc ions [11]. The binding curves obtained by MST and ITC show clear plateaus indicating the saturation of a specific copper binding site, and ruling out the possibility of an unspecific binding for which binding would increase linearly with the copper concentration (Figure 1). It was observed that while H7 has a *K*_D_ ranging between 700–100 nM, M7 displays a 20–50-fold lower affinity for copper, in line with a significantly reduced ROS generation and with the lack of in vivo toxicity of non-amyloidogenic LC.

To identify the residues directly involved in copper binding, the only two histidine residues present in the H7 sequence were mutated to alanine. The resulting H7-H188A/H197A displays the same affinity for copper as wild-type H7, thus demonstrating that histidine residues are not involved in copper binding. In line with these data, H_2_O_2_ generation by H7-H188A/H197A is almost identical compared to H7 wild-type.

Although the His-to-Ala substitutions did not abolish Cu^2+^binding, the Tm value shifts in the melting curves of the mutant allowed us to associate the lower and a higher midpoint temperatures to the serial unfolding of the V_L_ and C_L_ domains, respectively. While the C_L_ domain is destabilized in the H7-H188A/H197A mutant, the V_L_ domain is stabilized by its interaction with copper. The observed copper-induced stabilization of H7-V_L_ is in contrast to a previous study, in which copper binding destabilized the V_L_ domain of the LC 6αJL2-R24G, thus accelerating fibril formation [47]. We noted that the unique sequence of each individual LC V_L_-domain generated distinct molecular features that could lead to the observed (de-)stabilization differences caused by Cu binding. Thus, our data helped to localize the copper binding site in the V_L_ domain, but the specific binding residues remain to be determined.

Surprisingly, the two histidine mutations present in H7-H188A/H197A abrogate the proteotoxic effect of H7 in *C. elegans*. Fold stability is considered a relevant property in determining the toxicity and amyloid aggregation propensity; however, whether the reduced toxicity of the mutant is related to the decreased C_L_ stability, or the more cooperative fold, remains to be determined.

The soluble toxicity of amyloidogenic LCs is well supported by clinical observations [5,11,16,25,45], and experimentally proven using cellular approaches as well as animal models, such as *C. elegans* and zebrafish [48]. Several lines of evidence indicate that toxicity depends on specific molecular traits [10,11,25]

Although the pathogenic mechanisms responsible for the proteotoxic effect of amyloidogenic LCs remain to be fully elucidated, the findings in *C. elegans* suggest that it may imply multiple factors, from the oxidative and structural damage directly induced by ROS to the consequent activation of the intracellular signaling pathway aimed at repairing or protecting the damage [11].

Our previous and current investigations provide evidence that specifically bound Cu^2+^ enhances the toxicity of amyloidogenic LCs. However, the properties of the H7–H188A/H197A mutant further confirm that proteotoxicity, as amyloidogenicity, results from a complex interplay between several intrinsic molecular properties.

## 4. Materials and Methods

### 4.1. Recombinant LC Production and Purification

Recombinant LCs were produced according to [49]. Briefly, heterologous proteins were produced as inclusion bodies (IBs), and subjected to a refolding procedure, and refolded LCs were isolated by anion exchange (IEX) and size exclusion chromatography (SEC). In more detail, H7 and M7 were expressed in BL 21 DE3 *E. coli* from pASK-IBA33 plus and pET21b plasmids after induction with 0.43 μM anhydrotetracycline and 0.5 mM IPTG, respectively, for four hours at 37 °C. After IB isolation and refolding (10 mM Tris-HCl pH 8.0, 1 mM EDTA, 1 mM PMSF with 5 mM reduced L-glutathione and 0.5 mM oxidized L-glutathione), LCs were bound to Q SepharoseR Fast Flow 16/10 (GE Healthcare) equilibrated in 10 mM Tris-HCl pH 8.0. Refolded LCs eluted at a NaCl concentration of 160–240 mM during the applied 0–1M NaCl-gradient. LC dimer populations were isolated from the SEC column Superdex 200 10/300 GL SEC equilibrated in 10 mM Tris-HCl pH 7.5 at the retention volumes of ~14 mL, indicating the expected dimer structure.

### 4.2. Microscale Thermophoresis (MST)

Protein interaction studies were performed using MicroScale Thermophoresis (MST) according to [50,51]. Light chains were labeled using labeling kit amine-reactive NT-647-NHS fluorescent dye (Monolith NT™ Protein Labelling Kit RED-NHS, NanoTemper^®^ Technologies GmbH, München, Germany) according to the manual. Ions were titrated in 1:1 dilution starting from the highest final concentrations. Labeled light chain was added to obtain a final concentration of 60 nM. Light chains were exchanged to 10 mM TRIS-HCl pH 7.5 on Superdex 200 10/300 GL, and measurements were performed using standard-treated glass capillaries and the instrument Monolith NT.115 (Nano-Temper Technologies). The IR laser-power was set to 20%, the laser on-and-off times were set at 20 s and 5 s, respectively. The amplitude normalized data ΔF_norm_ of each dataset were averaged from the measurements shown in Appendix A and plotted against the concentration of the unlabeled ligand on a logarithmic scale. The binding data were analyzed using MO affinity analysis software, as provided by the manufacturer.

### 4.3. Isothermal Titration Calorimetry (ITC)

Light chains were exchanged to 10 mM TRIS-HCl pH 7.5. CuCl_2_ was dissolved in the same buffer used for SEC at a stock concentration of 10 mM. For the EDTA control experiments, the light chains and copper were made to 10 mM TRIS HCl 10 mM EDTA pH 7.5 following the same procedure. ITC measurements were performed using the Microcal PEAQ ITC calorimeter (Malvern). The cell temperature was set to 25 °C and the syringe stirring speed to 750 rpm. Before each experiment, the light chain and ions were loaded into the cell and syringe at concentrations of 20–35 μM and 200–600 μM, respectively. Except for the first injection with a volume of 0.4 uL, all the following injections were performed with a volume of 2 mL and a waiting time in-between each injection of 150 s. The data and binding parameters were pre-analyzed using the MicroCal PeakITC software (Malvern). The final analysis and global fits were performed using NITPIC and Sedphat, respectively [52,53].

### 4.4. Circular Dichroism and Tycho Measurements

Circular dichroism experiments, in the far- and near-UV regions, were carried out on a J-810 spectropolarimeter (JASCO Corp., Tokyo, Japan) equipped with a Peltier system for temperature control. All experiments were carried out in 10 mM TRIS- HCl pH 7.5. For the experiments in the far-UV region, the LCs were loaded at a concentration of 0.2 mg/mL into cuvettes with a pathlength of 0.1 cm. Spectra were recorded from 260 to 190 nm. Temperature ramps were applied from 20–80 °C with a slope of 60 °C/hour and monitored at 202 nm. For the near-UV region, the LCs were loaded at a concentration of 1 mg/mL into cuvettes with a pathlength of 1 cm. Spectra were recorded from 250 to 350 nm. Temperature ramps (as above) were monitored at 288 nm. Melting temperatures (Tm) were calculated as the first-derivative minima of the melting curves. The smoothed CD spectra and T-ramp curves were obtained by applying a Savitzky–Golay filter (Appendix A). The samples for nanoDSF protein melting curves were taken in capillaries specific for the Tycho NT1.6 instrument, immediately before or after the ITC measurements.

### 4.5. Data Integration and Figure Preparation

Pre-analyzed data were exported for data integration and visualization in R, mainly using tidyverse and ggplot packages [54,55,56,57]. The structural figure was rendered in PyMOL (Schrödinger, 2010).

### 4.6. Effect of LCs on C. elegans’ Pharyngeal Behaviour

The toxic effect of LC was evaluated, as previously described [10,11]. Briefly, Bristol N2 nematodes (Caenorhabditis elegans Genetic Center (CGC), University of Minnesota, Minneapolis, MN) were propagated at 20 °C on solid nematode growth medium (NGM) seeded with *Escherichia coli* OP50 (CGC) for food. Worms were incubated with 2.5–250 µg/mL H7 WT, mutated H7 or M7 (100 worms/100 µL) in 10 mM phosphate-buffered saline (PBS), pH 7.4. Ten mM PBS, pH 7.4, was administered as a negative control (vehicle). After 2 h of incubation on orbital shaking, worms were transferred onto NGM plates seeded with OP50 *E. coli* and the pharyngeal pumping rate was scored 24 h later. Experiments were also performed by feeding worms for 2 h with 10 or 100 µg/mL of LC alone or with 50 µM copper (II) (from CuCl_2_) in metal-free water. This dose of copper is compatible with viability of the worms [11].

### 4.7. Hydrogen Peroxide Determination

The amount of H_2_O_2_ generated by H7-wt, mutated H7 or M7 proteins (50 µg/mL in 10 mM PBS, pH 7.4) was determined before and at different times after incubation at 37 °C, using the Amplex^®^ Red Hydrogen Peroxide/Peroxidase Assay Kit (Catalog no. A22188, Thermo Fisher, Milan, Italy).

### 4.8. Statistical Analysis

The data were analyzed using GraphPad Prism 8.0 software (GraphPad Software, CA) by an independent Student’s *t*-test, one-way analysis of variance, and Bonferroni’s *post hoc* test. The values of IC50 were determined using Prism version 8.0 for Windows (GraphPad Software). A *p*-value < 0.05 was considered statistically significant.

## Figures and Tables

**Figure 1 ijms-23-00950-f001:**
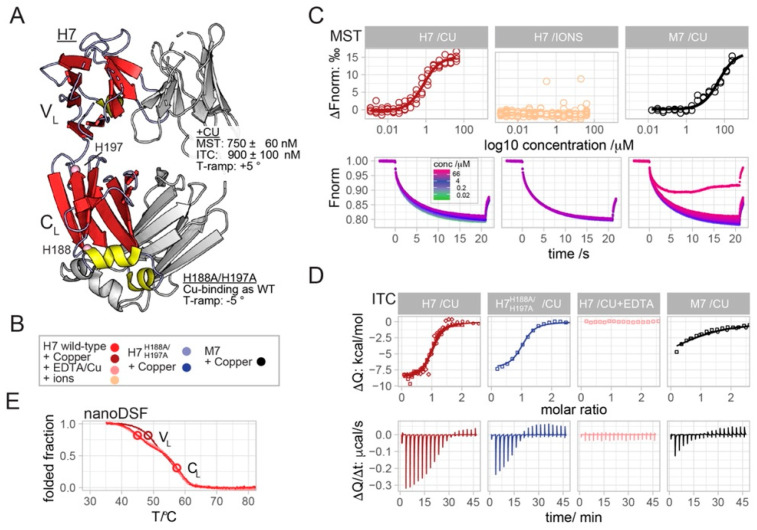
Combined MST, ITC and thermal unfolding demonstrate low micromolar binding affinity of copper to the V_L_ domain of H7. (**A**) The crystal structure of H7 (PDB: 5MUH) exemplifies the typical two-domain architecture and dimer interface observed for most LC structures [28]. Both the V_L_ and C_L_ immunoglobulin domains contribute to the dimer interface. The strands, helices and loops of one monomer are shown in red, yellow and grey. The second subunit is shown in grey. The positions of H197 and H188 are highlighted. (**B**) The color legend used throughout the manuscript unless stated otherwise. H7, H7-H188A/H197A and M7 were colored in tones of red, blue and black, respectively. (**C**,**D**) Selected MST time-traces and the baseline-corrected ITC thermograms are shown in the panels below the final MST titration curves and ITC binding isotherms. The complete set of MST and ITC raw data is shown in Appendix A. MST time-traces are color-coded on a log2-concentration scale. (**C**) The MST binding curve averaged from four Cu^2+^-to-H7 titration experiments (red) reveals a sigmoidal line-shape with an ΔF_norm_ amplitude of 15‰ and a fitted *K*_D_ value of 750 ± 60 nM. The titration curves of Ca^2+^, Fe^2+^, Mg^2+^, Na^+^ and Zn^2+^ to H7 over similar concentration ranges are summarized in the “ions” panel (light orange). The associated MST-TT of a single Mg^2+^ titration is shown below, all other raw data are shown in Appendix A. The titration of Cu^2+^ to M7-wt (black) yielded a binding curve with an ΔF_norm_ amplitude of 15‰ and a fitted *K*_D_ value 50 ± 15 μM, derived from two experiments. At the highest Cu^2+^ concentrations, we noticed wavy MST time-traces indicative for Cu^2+^ induced M7 aggregation, which were not included in the final analysis. (**D**) Offset-corrected ITC-binding isotherms demonstrated sub-micromolar affinity for the binding of Cu^2+^ to H7 (red) with *K*_D_ and binding the enthalpy (ΔH) values of 900 ± 100 nM and −8.7 ±  0.1 kcal/mole, respectively, as derived from global fits to four experiments. The substitution of His-188 and H-197 to Ala residues did not significantly alter the binding, yielding fitted *K*_D_ and ΔH values of 900 ±  300 nM and −7.6 ±  0.8 kcal/mole (blue). Chelation of Cu^2+^ by molar excess of EDTA abolished binding (dark red). The binding isotherm derived from two experiments for binding of Cu^2+^ to M7 (black) was fit with *K*_D_ and enthalpy (ΔH) values of 16 ± 6 μM and 6 ±  1 kcal/mole. (**E**) Thermal unfolding monitored at a 350/330 nm fluorescence ratio in nanoDSF revealed two melting temperatures (T_m_) for the unfolding of H7. The binding of copper (dark red) shifted the first Tm value from 45 to 48.3 °C. The thermal melting of H7 in the presence of EDTA (pink curve) yielded a profile almost identical to H7 in the absence of copper (light red).

**Figure 2 ijms-23-00950-f002:**
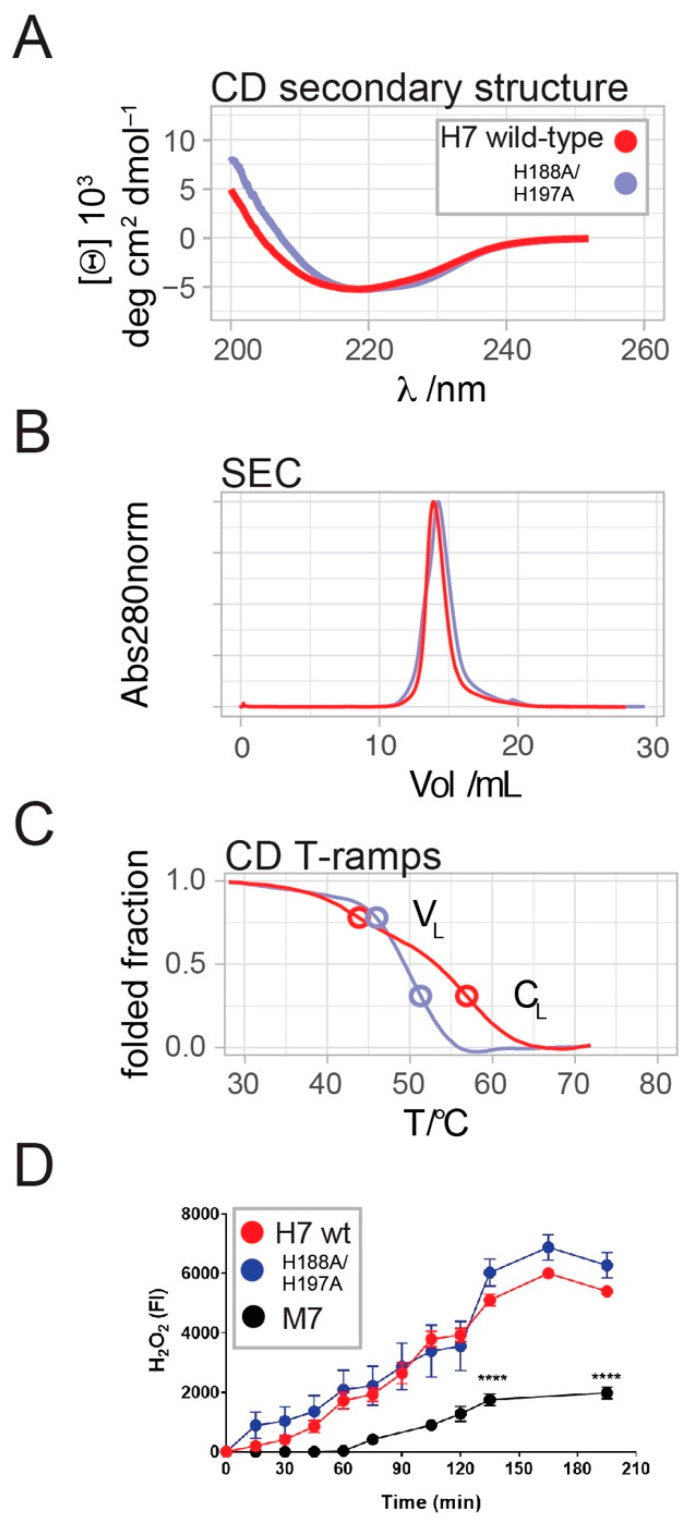
Mutated H7 has a similar overall structure but destabilized C_L_ domain compared to WT, and its capacity to produce H_2_O_2_ is unaltered. (**A**) CD spectra recorded for H7 (red) and H7-H188A/H197A (light blue) indicated minimal differences below 210 nm that are possibly caused by minimal loss of the strand and/or helix secondary structure in the C_L_ domain. (**B**) H7 and H7-H188A/H197A eluted from the analytical SEC column Superdex 200 10/300 GL at almost identical retention volumes of 14 mL, corresponding to an apparent MW of 44 kDa. (**C**) Comparative thermal melting of H7 and H7-H188A/H197A monitored at 202 nm in CD revealed that the 2nd inflection point was shifted from 57 to 51.5 °C. This destabilizing effect of the His-to-Ala substitutions in the C_L_ domain allowed us to assign the 1st and 2nd inflection points to the V_L_ and -C_L_ domains, respectively. Although the melting curve of H7-H188A/H197A is more cooperative with a single apparent transition, the hypothetical inflection points of the two domains based on the wild-type profile are indicated. (**D**) H_2_O_2_ produced by 50 µg/mL H7, H7-H188A/H197A and M7, incubated at different times at 37 °C in 10 mM PBS, pH 7.4. The mean ± SD of fluorescence intensity (FI), n = 6. **** *p* < 0.0001 vs. H7 WT and mutated H7, one-way Anova and Bonferroni’s *post hoc* test.

**Figure 3 ijms-23-00950-f003:**
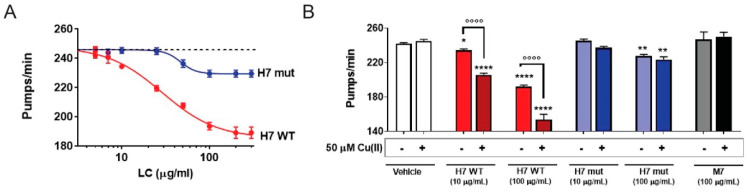
H7 mutant with destabilized C_L_ domain is less toxic to *C. elegans*. (**A**) Dose–response curves reveal the diminished toxicity of H7-H188A/H197A (H7 mut) compared to H7 (WT), as higher concentrations are required to inhibit the pumping rate of worms. Worms were fed for 2 h with different concentrations of WT or mutated H7 suspended in 10 mM PBS, pH 7.4, and the pharyngeal pumping was scored 24 h after the administration. Control worms received vehicle alone (dotted line). Each value is the mean ± SE, n = 30. IC_50_ was 28.9 and 46.8 µg/mL for WT and mutated H7, respectively, *p* < 0.01, Student’s *t*-test. (**B**) Worms were fed for 2 h with 10 or 100 µg/mL H7 or H7-H188A/H197A, or 100 µg/mL M7 dissolved in 10 mM PBS, pH 7.4, with or without 50 µM Cu (II). Control worms received 10 mM PBS, pH 7.4 with or without 50 µM Cu (II) (vehicle). Pharyngeal pumping was determined 24 h after the administration. Each value is the mean ± SE, n = 20. * *p* < 0.05, ** *p* < 0.01 and **** *p* < 0.001 vs. the corresponding vehicle, °°°° *p* < 0.001, one-way ANOVA and Bonferroni’s *post hoc* test.

## Data Availability

Data is contained within the article or Appendix A.

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
