# Peer review of "Cu(II) Binding Increases the Soluble Toxicity of Amyloidogenic Light Chains"

_ijms, 2022, doi:10.3390/ijms23020950_

Round 1

Reviewer 1 Report

Title: Cu(II) binding increases soluble toxicity of amyloidogenic light chains   In this work authors have studied the effect of copper binding on the soluble toxicity of amyloidogenic light chains. Several methods have been used to conclude the results. Overall, the work is interesting and has the potential to be published in International Journal of Molecular Sciences. However, I am concerned about the working pH in case of 'Recombinant LC production and purification' and 'Microscale thermophoresis'. What is the reason behind selecting pH 8.0? Furthermore, the first paragraph of the abstract can be omitted because all these detailed information are given in introduction section.   

Author Response

Referee 1

Comments and Suggestions for Authors

Title: Cu(II) binding increases soluble toxicity of amyloidogenic light chains   In this work authors have studied the effect of copper binding on the soluble toxicity of amyloidogenic light chains. Several methods have been used to conclude the results. Overall, the work is interesting and has the potential to be published in International Journal of Molecular Sciences.

- However, I am concerned about the working pH in case of 'Recombinant LC production and purification' and 'Microscale thermophoresis'. What is the reason behind selecting pH 8.0?

We thank for this comment. While the pH 8.0 is the optimised pH for refolding and purification, all binding experiments have been performed at pH 7.5. Indeed, the sentence describing the MST experiments at pH 8.0 is a mistake and the Methods section has been amended accordingly.

- Furthermore, the first paragraph of the abstract can be omitted because all these detailed information are given in introduction section.   

According to Referee 1 comment, we shortened the introductory paragraph of the abstract, however for sake of non-expert readers we kept some basic information about AL amyloidosis in the abstract.

Reviewer 2 Report

The manuscript “Cu(II) binding increases soluble toxicity of amyloidogenic light 2 chains” by Rosaria Russo et al. aims at understanding the role of copper ions ofn the toxicity of amyloidogenic light 2 chains. The authors addressed a highly important question on the metal induced toxicity. To support their data, the authors used a combination of f MST, ITC and thermal melting to demonstrate specific binding of Cu2+ ions to protein variable domain. The authors find that the copper binding is just one of the several biochemical traits contributing to LC soluble toxicity in vivo.

This is a very interesting and timely paper and I have thoroughly enjoyed reading reviewing this manuscript. The topic is of high importance and the effect of metal ions on protein toxicity have gained tremendous attention in recent years. Work data presented in the paper will increase our knowledge about this important aspect of protein toxicity and aggregation. The experiments are well designed and the results are clean. The approach is simple and elegant, and easy to follow and the interpretation is convincing. The paper is also clearly written and accessible to a broad audience, and the figures are clear and helpful. I am rather enthusiastic about this paper and supportive of its publication. I only offer some minor suggestions to improve readability and enhance the message of the paper (adopting them is optional).

In sum, I very much enjoyed this very interesting paper and I am positive about its publication.

Minor issues:

  1. Figure 2. CD spectra does not look smooth. Is it not average of atleast triplicates?
  2. It is not clear which residues are involved in Copper binding? Authors should shed some light on this.
  3. The copper induced effect is result of specific direct binding or its non-specific effect?

Author Response

Referee 2

The manuscript “Cu(II) binding increases soluble toxicity of amyloidogenic light 2 chains” by Rosaria Russo et al. aims at understanding the role of copper ions ofn the toxicity of amyloidogenic light 2 chains. The authors addressed a highly important question on the metal induced toxicity. To support their data, the authors used a combination of f MST, ITC and thermal melting to demonstrate specific binding of Cu2+ ions to protein variable domain. The authors find that the copper binding is just one of the several biochemical traits contributing to LC soluble toxicity in vivo.

This is a very interesting and timely paper and I have thoroughly enjoyed reading reviewing this manuscript. The topic is of high importance and the effect of metal ions on protein toxicity have gained tremendous attention in recent years. Work data presented in the paper will increase our knowledge about this important aspect of protein toxicity and aggregation. The experiments are well designed and the results are clean. The approach is simple and elegant, and easy to follow and the interpretation is convincing. The paper is also clearly written and accessible to a broad audience, and the figures are clear and helpful. I am rather enthusiastic about this paper and supportive of its publication. I only offer some minor suggestions to improve readability and enhance the message of the paper (adopting them is optional).

In sum, I very much enjoyed this very interesting paper and I am positive about its publication.

Minor issues:

Figure 2. CD spectra does not look smooth. Is it not average of at least triplicates?

According to Referee 2 suggestion we smoothed the curves shown in fig 2A and 2C, thus CD spectra and unfolding curves are more easily readable and comparable. We added here a comparison (not for publication) of smoothed and unsmoothed CD data in order to show the reliability of the smoothed data.

- It is not clear which residues are involved in Copper binding? Authors should shed some light on this.

The copper induced effect is result of specific direct binding or its non-specific effect?

Referee 2 raises crucial pojnts. Based on our data we identified that Copper binds specifically to the variable domain (VL). We could also rule out Met, Cys, and His residues which are typically involved in copper binding, However, we did not identify the exact binding site which would require substantial additional crystallography or NMR structural investigation. We believe that such efforts are beyond the scope of this work, and could be addressed in future studies. A sentence in the discussion (pg 12) has been added to clarify this point.

The binding curves obtained by MST and by ITC display a plateau indicating the saturation of a specific binding site and rule out non-specific binding which would lead to curve with an ever-increasing signal along with increasing concentration of copper ions. At page 11 in the first paragraph of the Discussion, we stated more clearly such concept.
